# Diversification and recurrent adaptation of the synaptonemal complex in *Drosophila*

**Rana Zakerzade**[1,2☉], **Ching-Ho Chang**[3☉], **Kamalakar Chatla**[4], **Ananya Krishnapura**[4], **Samuel P. Appiah**[4], **Jacki Zhang**[4], **Robert L. Unckless**[5], **Justin P. Blumenstiel**[6], **Doris Bachtrog**[4], **Kevin H-C. Wei**[1,2,4]*

**1** Department of Zoology, University of British Columbia, Vancouver, British Columbia, Canada, **2** Life Sciences Institute, University of British Columbia, Vancouver British Columbia, Canada, **3** Basic Sciences Division, Fred Hutch Cancer Center, Seattle, Washington, United States of America, **4** Department of Integrative Biology, University of California Berkeley, Berkeley, California, United States of America, **5** Department of Molecular Biosciences, University of Kansas, Lawrence, Kansas, United States of America, **6** Department of Ecology and Evolutionary Biology, University of Kansas, Lawrence, Kansas, United States of America

☉ These authors contributed equally to this work.

* wei.kevin@ubc.ca

**Data Availability Statement:** All generated raw sequences have been deposited onto the SRA under Bioproject PRJNA1030345. Relevant intermediate datasets and files have been

## Abstract

The synaptonemal complex (SC) is a protein-rich structure essential for meiotic recombination and faithful chromosome segregation. Acting like a zipper to paired homologous chromosomes during early prophase I, the complex is a symmetrical structure where central elements are connected on two sides by the transverse filaments to the chromatin-anchoring lateral elements. Despite being found in most major eukaryotic taxa implying a deeply conserved evolutionary origin, several components of the complex exhibit unusually high rates of sequence turnover. This is puzzlingly exemplified by the SC of *Drosophila*, where the central elements and transverse filaments display no identifiable homologs outside of the genus. Here, we exhaustively examine the evolutionary history of the SC in *Drosophila* taking a comparative phylogenomic approach with high species density to circumvent obscured homology due to rapid sequence evolution. Contrasting starkly against other genes involved in meiotic chromosome pairing, SC genes show significantly elevated rates of coding evolution due to a combination of relaxed constraint and recurrent, widespread positive selection. In particular, the central element *cona* and transverse filament *c(3)G* have diversified through tandem and retro-duplications, repeatedly generating paralogs with novel germline activity. In a striking case of molecular convergence, *c(3)G* paralogs that independently arose in distant lineages evolved under positive selection to have convergent truncations to the protein termini and elevated testes expression. Surprisingly, the expression of SC genes in the germline is prone to change suggesting recurrent regulatory evolution which, in many species, resulted in high testes expression even though *Drosophila* males are achiasmic. Overall, our study recapitulates the poor conservation of SC components, and further uncovers that the lack of conservation extends to other modalities including copy number, genomic locale, and germline regulation. Considering the elevated testes expression in many *Drosophila* species and the common ancestor, we suggest that the

deposited on Dryad (https://doi.org/10.5061/dryad.00000008q).

**Funding:** This work was funded by the National Science and Engineering Research Council Discovery Grant (RGPIN-2023-05390), National Institute of Health K99 Pathway to Independence Award (K99GM137041-01A1), and Michael Smith Health Research BC Scholar Award (SCH-2024-03818) to KHCW; and the Damon Runyon Fellowship Award (DRG 2438-21) to CHC The funders had no role in study design, data collection and analysis, decision to publish, or preparation of the manuscript.

**Competing interests:** The authors have declared that no competing interests exist.

activity of SC genes in the male germline, while still poorly understood, may be a prime target of constant evolutionary pressures driving repeated adaptations and innovations.

## Author summary

The synaptonemal complex (SC) is essential for meiotic recombination and faithful chromosome segregation across eukaryotes, yet components of the SC are often poorly conserved. Here, we show that across the *Drosophila* phylogeny, SC genes have evolved under recurrent positive selection, resulting in orthologs and paralogs often with barely recognizable amino acid sequences. This is partly driven by duplications repeatedly generating paralogs, many of which appear to have adopted novel germline expression patterns, often highly active in the testes. While most SC genes are thought to be dispensable in the male germline of *Drosophila* where meiotic recombination does not occur, elevated testes expression independently emerged in different lineages and appears to be the norm across the genus and likely the ancestral state. Unexpectedly, SC expression in ovaries is also poorly conserved, revealing recurrent regulatory turnover. We suggest that the evolutionary lability of SC genes in *Drosophila* is likely a repeated source of functional diversifications and innovations in the germline.

## Introduction

Meiotic recombination, the exchange of non-sister, homologous chromosomes through physical crossovers, is an essential genetic mechanism universal to sexually reproducing eukaryotes. It allows for the shuffling of homologous alleles generating novel allelic combinations. This is necessary for maintaining nucleotide diversity and efficacy of selection; without it, chromosomes (like on the non-recombining, degenerate Y or W chromosomes) will irreversibly accumulate deleterious mutations ultimately leading populations to go extinct. At the cellular level, meiotic pairing, synapsis, and resolution of double strand breaks into crossovers are critical for stabilizing meiotic bivalents as failure is typically associated with skyrocketing aneuploidy rates [1,2]. Therefore, recombination is a crucial genetic process that is necessary for reproductive fitness and long-term species survival.

Despite the critical functionality of recombination and the deep conservation across eukaryotes, aspects of this fundamental genetic mechanism are surprisingly prone to change. Recombination rate has been repeatedly shown to vary drastically between closely related species [3]. Adaptive explanations typically invoke changing environmental (e.g. temperature [4]) or genomic conditions (e.g. repeat content [5]) requiring commensurate shifts in recombination rate to maintain fitness optima [6,7]. Others have suggested intragenomic conflicts with selfish elements [8] or sexual conflict creating unstable equilibria for optimal fitness [9,10]. However, some have argued that changes in recombination rate have little impact on fitness and rate changes are the byproduct of selection on other aspects of the meiotic processes [11]. Several key findings supporting the adaptive interpretation come from *Drosophila* as multiple genes in the pathways necessary for recombination show signatures of rapid evolution due to positive selection [12–15]. Moreover, because recombination is absent in *Drosophila* males and the machinery absent during spermatogenesis [16], sexual antagonism due to sex-specific optima of crossover rates is thought to be an unlikely driver of adaptive recombination evolution, at least in species with sex-specific achiasmy. Why recombination, an essential genetic mechanism, is prone to change and whether such changes are adaptive remain central questions in evolutionary genetics [17–19].

The paradox of poor conservation but crucial function is exemplified by the synaptonemal complex (SC), a crucial machinery necessary for meiotic recombination in plants, animals, and major lineages of fungi [20]. It is a protein complex that acts as zippers to tether homologs together along the chromosome axes during meiotic prophase I and forms train track-like structures which have been visualized under electron microscopy across eukaryotic taxa [21,22]. The SC is mirrored along a central axis composed of central element proteins that are tethered by the transverse filaments to lateral elements on two sides anchoring into chromatin (Fig 1A) [23]. This highly stereotypical configuration is found in baker's yeast, mice, flies, and plants, indicative of an evolutionary ancient structure. Yet, despite the deep evolutionary origin and functional necessity across wide eukaryotic domains, there are many examples of unexpected exceptions. At the extreme are recombining species such as the fission yeast that entirely forego the SC [24]. In another instance, the SC of *Caenorhabditis* has been reconfigured such that the transverse filament–typically a single gene in most SCs–is composed of at least four genes [25]. Therefore, parts of the SC appear to be curiously flexible in composition whereby different analogous but perhaps non-homologous pieces can be recruited and replaced [26].

Consistent with this flexibility, sequences of SC components are often poorly conserved at shorter evolutionary time scales [14,27]. In *Drosophila*, positive selection appears to repeatedly drive the sequence evolution of the SC, which is composed of the central elements *corona* (*cona*) [28] and *corolla* [29], the transverse filament *c(3)G* [30], and the lateral elements *orientation disruptor* (*ord*) [31] and *c(2)M* [32]. Previously, orthologs of the central region components, *corolla*, *cona*, and *c(3)G*, could not be found outside of the *Drosophila* genus [14] either reflecting divergence so extensive that orthology is no longer recognizable, or novel acquisitions of SC components. Flexibility in SC composition may explain how these molecular transitions are possible without major fitness impacts, but cannot account for why SC genes appear to be evolving under recurrent adaptation. The recent explosion of high quality *Drosophila* species genome assemblies [33–40] offers a unique opportunity to understand the genetic and evolutionary mechanisms driving the puzzlingly rapid divergence of SC genes. Here, we systematically revisit the evolution of the SC in *Drosophila* by examining the genomes and transcriptomes of 48 species scattered across the entire *Drosophila* phylogeny, with dense representation from three key species groups (*melanogaster*, *obscura*, and *immigrans*). In our exhaustive analyses, we uncovered frequent duplications of several SC components generating paralogs with likely novel functions, in addition to repeated sequence evolution under positive selection. Further, we revealed unexpectedly high rates of expression divergence and regulatory turnover in not just the ovary but also the male germline, where SC genes are thought to have no function. In fact, testes-biased expression of SC genes appears to be the norm, and likely the ancestral state, suggesting SC components have crucial function in male germline, despite the absence of male recombination. Altogether our study revealed a highly dynamic evolutionary history with repeated bouts of copy number, sequence, and regulatory evolution that contribute to the overall poor conservation of SC genes. Further, the surprising transcriptional activity of SC genes in the male germline raises new possibilities for functions of SC genes unrelated to recombination under repeated directional selection in addition to their roles in chiasmate meiosis in the female germline.

## Results

### Poor sequence conservation and frequent duplications of components of the SC

To identify *Drosophila* SC homologs we elected to focus on only species with high quality genome assemblies with either available annotations and/or RNA-seq data (S1 Table). In

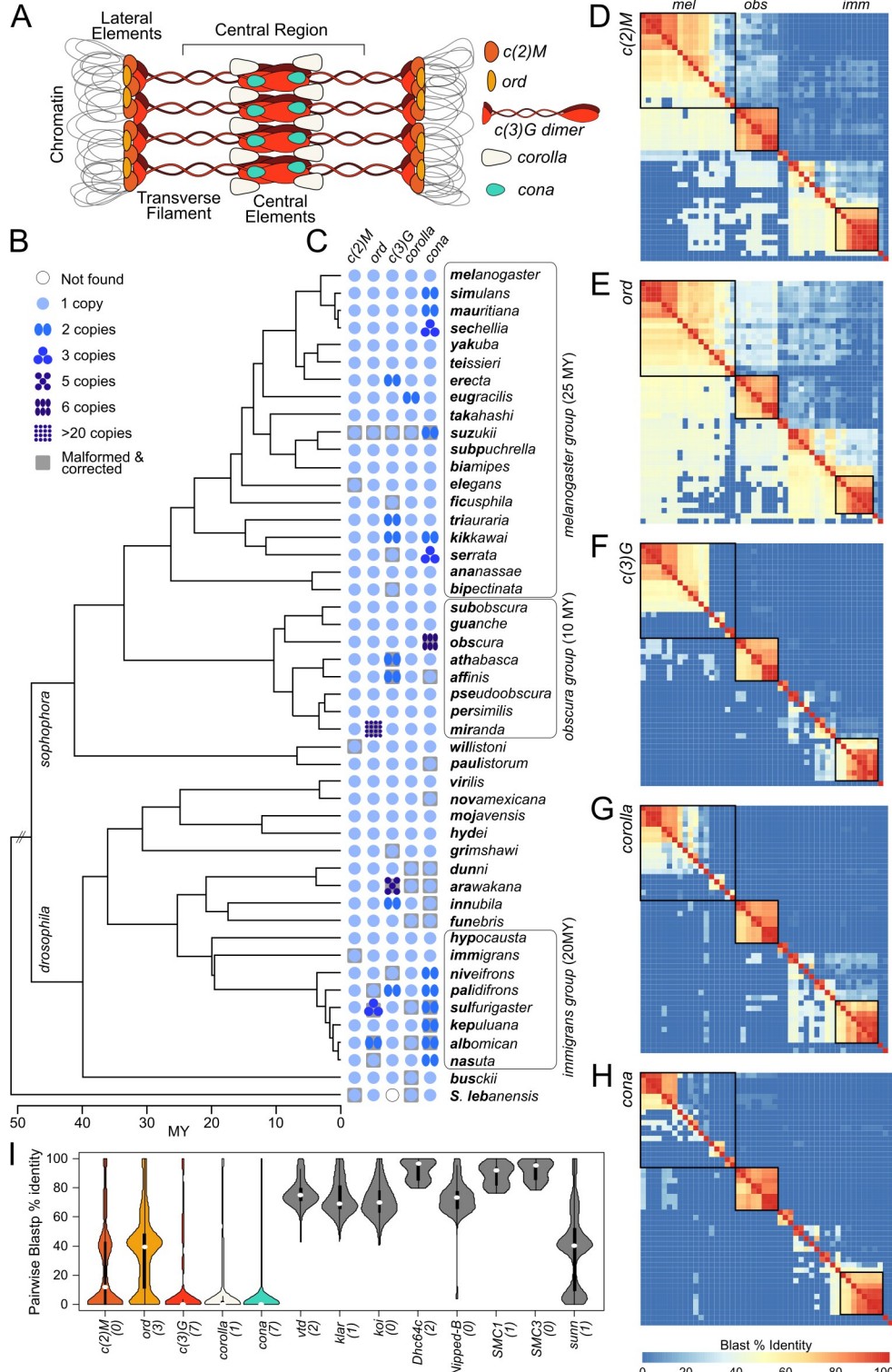

**Fig 1. Sequence conservation, or the lack thereof, of synaptonemal complex components across the *Drosophila* genus.** A. Cartoon diagram of the *Drosophila* SC and its primary constituents. B. Phylogenetic relationships of the 48 species used in this study. Bold letters in the names denote species shorthands. Species dense groups are labeled and boxed. C. Presence, copy number, and absence of SC components across the phylogeny. The number of blue circles indicated the copy number. D-H. Pairwise blast sequence alignments between orthologs from species across the genus. Alignments above the diagonal are from nucleotide blasts of the CDS sequences using blastn. Alignments below the

diagonal are from protein blasts of the amino acid sequence using blastp. % blast identity is the % coverage of blast alignments multiplied by the % sequence identity, summed across the gene. I. Distribution of pairwise blastp % identity (excluding self-blast) across the genus for SC (colored) and genes (gray) involved in meiotic pairing during early prophase. Numbers in parentheses are the number of lineages with duplicates.

addition, we strategically generated highly contiguous assemblies of two additional species (*D. hypocausta* and *D. niveifrons*, belonging to the immigrans group; S2 Table), and testes and ovaries RNA-seq of eight species (*D. subobscura*, *D. arawakana*, *D. dunni*, *D. innubila*, *D. funebris*, *D. immigrans*, *D. hypocausta*, and *D. niveifrons*) to either annotate previously unannotated genomes or to refine previous annotations (S1 Table). Altogether, we compiled a total of 47 species spanning the two major arms of the *Drosophila* genus (the *Sophophora* and *Drosophila* subgenera), with three species groups particularly well-represented (*melanogaster*, *obscura*, and *immigrans* groups) (Fig 1B) and the outgroup species *Scaptopdrosophila lebanonensis*.

Using a multi-step reciprocal best blast hit approach, we sought to identify orthologs and paralogs across the genus (Fig 1B and 1C; Material and methods). However, we found that the gene structures are frequently malformed regardless of the source of the annotation (publicly available, or our ones we generated). SC components are often mis-annotated as truncated or chimeric gene products or entirely missing in the annotation (S3 Table; for examples see S1 Fig), likely due to the combination of exacerbating factors such as poor sequence conservation, frequent presence of tandem duplicates, low RNA-seq reads, and in some cases assembly errors. To ensure proper sequence alignments, we therefore curated all SC genes and manually re-annotated all erroneous ones ensuring at a minimum, well-formed CDSs and intact full ORFs (S1 and S2 Figs; see Materials and Methods). We note that because *cona* is a short gene with few exons, we hand-annotated its orthologs in 8 additional species (see below).

For the lateral elements *c(2)M* and *ord*, sequence homology is somewhat preserved across the genus, although homology is not always detectable between distant species using Blast (Fig 1D and 1E However, for the central region genes (*c3G*, *cona*, *corolla*), DNA sequence homology quickly becomes unrecognizable outside of species groups, while weak protein homology is only occasionally detectable (Fig 1F–1H). Previously, Hemmer and Blumentiel 2018 identified SC orthologs in a subset of fly species [14]. Increased species and better annotations enabled us to identify orthologs previously missed (*cona* in *D. willistoni* and *corolla* in the outgroup) and resolve discrepant homology relationships (*cona* in the *Drosophila* subgenus, see below). To determine whether this lack of conservation is common to other genes involved in early meiotic progression, we curated and identified the homologs of eight genes necessary for meiotic chromosome pairing [23] across the *Drosophila* genus (S3 Fig). The poor sequence conservation of the SC starkly contrasts from these (Fig 1I): even the most conserved component of the SC, *ord*, shows significantly poorer sequence homology compared to the least conserved meiotic pairing gene, *sunn* which directly interacts with the SC (Fig 1I; Wilcoxon's Rank sum test p = 1.813e-05).

Furthermore, we uncovered multiple independent duplication events, with *c(2)M* being the only SC gene that remained single-copy across the genus. All SC paralogs were previously unaccounted for with the only exception being *ord* duplicates in *D. miranda*, which was identified to have rampantly amplified creating over 20 copies (S2 Fig) on the species' unique neo-sex chromosomes [41]. Of the poorly conserved components, *c(3)G* and *cona* in particular have recurrent copy number changes, having more than two copies in eight and thirteen species, respectively, as the result of independent duplications in at least seven lineages each. This propensity to duplicate is epitomized by the five *c(3)G* copies in *D. arawakana* and six *cona* copies in *D. obscura*.

## Complex history of repeated duplication and loss of paralogs

Based on protein trees of the SC components, we find that the current copy number distributions reflect at least 3, 7, 1, and 7 independent duplications of *ord*, *c(3)G*, *corolla*, and *cona*, respectively. The majority of paralogs are recent species-specific duplications resulting in short branch lengths (Figs 2A, 2B and S4 The genomic locations of the copies further reveal that tandem duplications account for the majority of the observed copies. For the transverse filament *c(3)G*, four of the seven duplication events are tandems (Fig 2A, 2C and 2D), three of which (including the five copies in *D. arawakana)* are recent and species-specific. In the lineage leading up to *D. athabasca* and *affinis*, an older tandem duplication generated duplicates of *c(3)G* and the neighboring gene *pon*, (Fig 2D); one of the copies which we designated *c(3)G2* is shorter, while showing poorer conservation and longer branch lengths between the orthologs, suggestive of elevated divergence (see below). Similarly, for *cona*, three of the seven duplication events are tandems, including the 6 copies in *D. obscura* (Figs 2B and S5).

Both *c(3)G* and *cona* experienced several instances of likely retroduplications. *cona* offers two clear examples of old events in the common ancestor of the *serrata* and *nasuta* subgroups leading to paralogs found on different chromosomes shared across all species in the subgroups (Fig 2B). *c(3)G*'s duplication history appears more convoluted but offers unique insight into its dynamic evolution. There are three independent non-tandem duplicates of *c(3)G* found in *D. kikkawai*, *triauraria*, and *innubila* (Fig 2A); the resulting paralogs are found on different chromosomal regions (Figs 2A and S6A) and have no neighboring homology to the original (Figs 2E bottom panel and S6B) suggesting retroduplication events instead of larger scale duplicated translocations. In the latter two species, the duplications are older events evidenced by their phylogenetic placements with long branches separating the paralogs, compared to the species-specific duplication in *D. kikkawai*. For *D. triauraria*, the duplication creating *c(3)G2* predated the split of the serrata species subgroup, but is no longer found in the derived lineages, indicating subsequent loss of the paralog. For *D. innubila*, the paralogs are found 180kb apart on the X chromosome (S5B Fig), and the phylogeny suggests that the duplication occurred after the split from *D. funebris*, although with low bootstrap support (Fig 2A). Interestingly, synteny information suggests this is not the true relationship; while one of the *c(3)G* copies is found in the same syntenic block shared with *D. funebris*, *arawakana*, and *dunni*, the other is in a separate synteny block shared with most species in the *Drosophila* subgenus and thus the ancestral copy (Figs 2A and S6C). This synteny pattern is therefore more parsimonious with an old duplication in the last common ancestor of the *dunni*, *quinaria*, and *funebris* species groups with the original copy lost in species other than *D. innubila*. Non-allelic gene conversion subsequently homogenized the duplicates in *D. innubila*, obscuring the true phylogenetic relationship.

In addition to the one *c(3)G* retroduplicate in *D. kikkawai*, we curiously identified numerous loci across the genome as 5' truncated homologs, none of which were annotated or have RNA-seq reads mapping (Fig 2E, top panel). These truncated and nonfunctional duplicates, along with the two loss events mentioned, raise the possibility that *c(3)G* experienced not only repeated duplications through transpositions, but also repeated pseudogenization events. A similar pattern of nonfunctional duplicates is also observed with *corolla* in *D. arawakana*; despite only one full length *corolla*, there are four adjacent tandem copies that lack the 5' exon and therefore likely non-functioning (S7 Fig). Further examining the syntenic relationships of the SC homologs, we surprisingly find that while the lateral elements have maintained the same local microsynteny showing a lack of gene movement, the central region genes have repeatedly relocated to different chromosomes, or different locations on the same chromosome even in lineages with stable copy number (Figs 2A,2B and S6A). The X is likely the

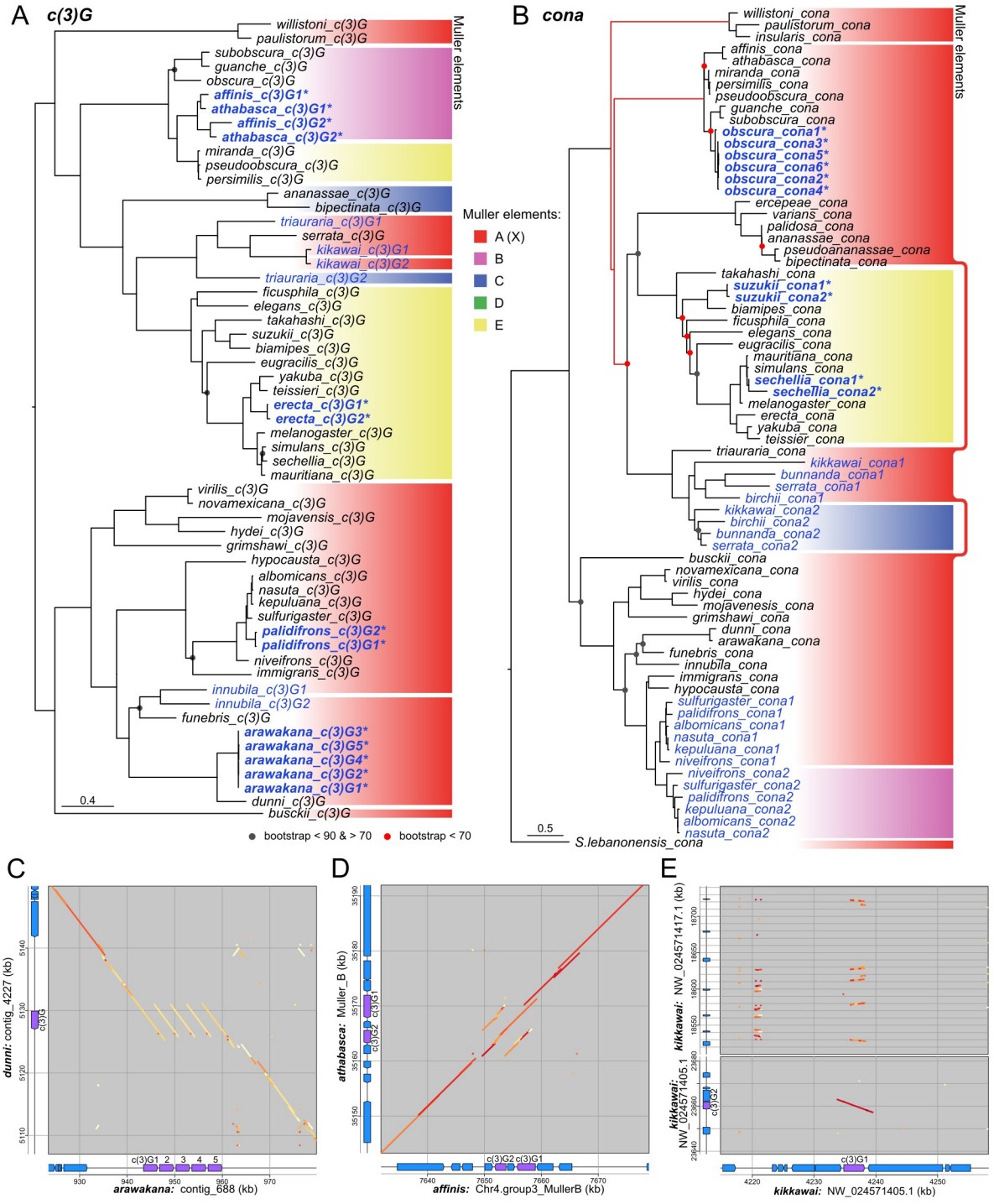

**Fig 2. Complex evolution history of synaptonemal complex genes.** A and B. Gene trees of *c(3)G* and *cona* orthologs and paralogs, respectively; nodes with poor bootstrap support are marked by circles. Duplicates are in blue and tandem duplicates have asterisks and in bold. For *cona*, some branches were adjusted (red) to align with the species tree. For other SC components see S4 Fig. Color blocks to the right represent different Muller elements on which the homologs reside. Separate blocks of the same color but not joined by lines represent homologs found in different locations on the same element. C-E. Dotplots showing synteny of genomic regions surrounding *c(3)G* between sister species and/or paralogs. The color of the dots represents the % sequence identity from blastn alignments with darker red reflecting higher identity. In the gene tracks of the displayed regions, *c(3)G* is in purple and other neighboring genes in blue. E. Lower panel shows the local homology of *c(3)G1* and retroduplicate *c(3)G2* in *D. kikkawai*; upper panel shows additional regions with multiple truncated duplicates (top).

ancestral home for all three but we documented eleven, six, and five inter- and intrachromosomal movements for *c(3)G*, *corolla*, and *cona*, respectively. Such recurrent movements likely through transpositions are highly unusual for flies. Despite frequent large scale rearrangements scrambling broad chromosome-scale synteny, chromosomal gene content and microsyteny in flies are largely stable [42]. Curiously, one relatively recent relocation occurred in the common ancestor of the *pseudoobscura* species subgroup, moving *c(3)G* from Muller B to an euchromatic repeat block on Muller E (S8 Fig), suggesting that such movements may be mediated by the instability of repetitive sequences, perhaps piggybacking off of transpositions of transposable elements. Since most of these movements no longer have extant paralogs, corresponding pseudogenization events were likely common. Therefore, even though most observable paralogs are young tandem duplicates, retroduplications and pseudogenization events have frequently occurred for *c(3)G*, *cona*, and even *corolla* which has few remnants of duplicates, thus accounting for the existence of many recent and species-specific copies in different locations but fewer old, shared duplicates.

## Evidence for functional diversification and germline activity of *c(3)G* and *cona* paralogs

Much like other transverse filaments, *c(3)G* has an extensive coiled-coil domain flanked by globular domains at the N- and C- termini connecting the central and lateral elements, respectively [20,43]. Despite the poor sequence conservation, we find that this canonical structure is conserved across the genus based on AlphaFold protein predictions (Fig 3A) [44,45] and coiled-coil predictions (S9 Fig) [46]. This unique evolutionary property of structural but not sequence conservation is also observed by Kursel et al. in *Caenorhabditis*, whereby central element genes have conserved coiled-coil domains and near invariant protein lengths, while

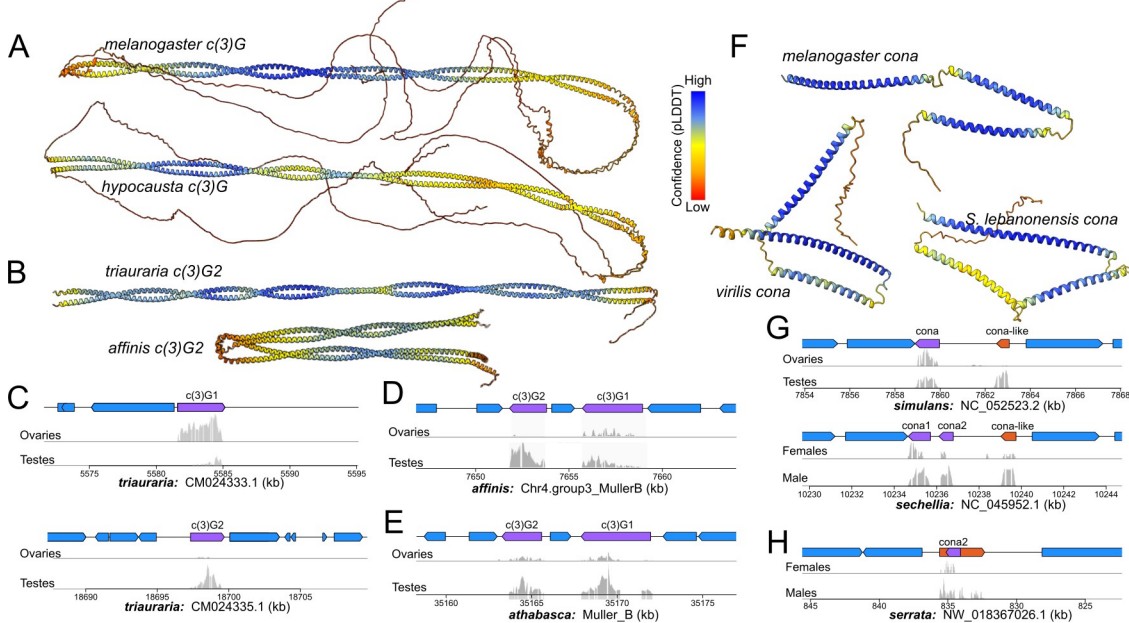

**Fig 3. Functional and structural evolution of c(3)G and cona.** A-B. AlphaFold structural prediction of full length c(3)G (A) and diverged paralogs (B). Color represents the confidence of structural prediction. C-E. Gene structure and gonad expression of c(3)G paralogs (purple genes) in D. triauraria (C), affinis (D) and athabasca (E). F. Alphafold of distant cona orthologs in Drosophila and outgroup species. G. Gene structure and gonad expression of cona paralogs in D. simulans and sechellia. Annotated lncRNA of gene with homology to cona is in orange in the gene tracks. G. Same as F but for D. serrata.

being some of the most divergent proteins [27]. In *D. athabasca*, *affinis*, and *triauraria*, while *c(3)G1*s produce longer proteins (690, 692, 830 AAs respectively) predicted to have the canonical structure, the paralogs *c(3)G2*s all produce notably shorter proteins (361, 395, and 319 AAs, respectively). Even though the *D. triauraria c(3)G2* and *D. athabasca/affinis c(3)G2* arose independently and are found on different chromosomes (Fig 2A), these paralogs share remarkably similar structural changes; the flanking globular domains which are necessary for attachment to the lateral and central elements [47] have been truncated, if not entirely absent, strongly suggesting that they no longer function as transverse filaments that can tether the SC (Fig 3B). Moreover, these paralogs are highly expressed in the testes but lowly expressed in the ovaries (Fig 3C–3E), incongruent with the expectation of female-specific meiotic function. Therefore, despite being independent duplications in lineages separated by over 35 million years, *c(3)G2* in *D. triauraria* and *D. athabasca/affinis* display similar structural and regulatory evolution, revealing molecular convergence for putative male germline activity. In a single nuclei RNA-seq dataset of *D. affinis* testes, we further find that while *c(3)G1* and *c(3)G2* are both testis-expressed, they are most active in different cell populations in the early germline (S10 Fig), further supporting functional divergence after duplication.

Similar to *c(3)G*, *cona* has maintained the same conserved tri-coil structure (Fig 3F), despite poor protein homology. Several *cona* duplicates also show properties that deviate from its characterized function in SC formation during female meiosis. At least two recent duplication events occurred within the *simulans* clade generating two upstream paralogs, one ancestral to the three *simulans* species while another found only in *D. sechellia* (Figs 2B and 3G). The *sechellia*-specific duplicate generates a complete but short ORF and likely protein coding, but the shared paralog only shows homology at the 3', lacks a complete ORF, and is annotated as a long non-coding RNA (Fig 3G). To evaluate whether this paralog, which we named *cona-like*, is transcriptionally active or pseudogenized, we examined RNA-seq data, and found high expression in the testes and males but low-to-no expression in ovaries or females across all *simulans* species (Fig 3G), suggesting testes function as a lncRNA. Adding to the intrigue, this is not the only instance of a *cona* paralog generating lncRNA. In the *serrata* group, the retroduplicate *cona2*, is shared across the species (Fig 2B), but only in *D. serrata* does it generate a lncRNA (Fig 2H). Unlike *cona-like* in the *simulans* clade, this paralog has a well-formed ORF and is expressed in both sexes, suggesting high protein coding potential. However, the lncRNA is anti-sense as confirmed with strand-specific RNA-seq (S11 Fig) and includes additional flanking sequences that only show expression in males. *cona2* likely generates a functional protein in females and ovaries but was incorporated in the anti-sense direction into lncRNA production in the testes of *D. serrata*. Altogether, these results suggest that both *c(3)G* and *cona* paralogs have repeatedly adopted germline activity in the testes unrelated to SC formation.

## coronetta (*conta*) is an ancient testes-expressed paralog of *cona*

Previously, Hemmer and Blumenstiel identified *GJ20698* –a gene producing a short peptide of 109 AA–in *D. virilis* as the *cona* ortholog using reciprocal best blast hit [14]. Although we found the orthologs of *GJ20698* across the *Drosophila* subgenus as well as the outgroup, none were reciprocal best hits to *sophophora cona*; in fact, they have no identifiable *sophophara* homologs at all. Instead, increased density of species enabled us to correctly identify *D. virilis's GJ16397* (*D. virilis's* second best hit to *sophophora cona*) which has orthologs across the *Drosophila* subgenus and *S. lebanonensis*, all of which are reciprocals best hits with *sophophora cona* (Figs 1F and 2B). The gene tree affirms that *sophophora cona* is more closely related to *GJ16397*, which we conclude to be the true *cona* ortholog (and used for all analyses). *GJ20698*, which we named *coronetta* (*conta*), appears to be a distant paralog, and, given its presence in

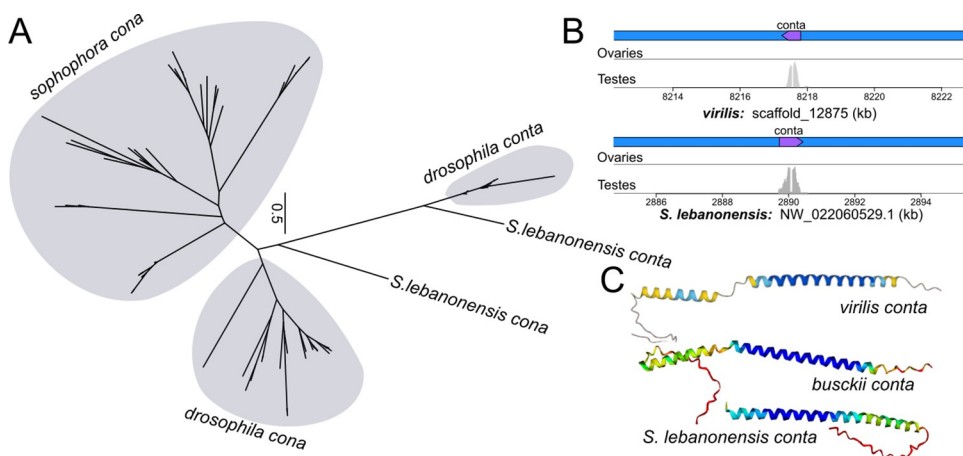

**Fig 4. Expression and structure of *conta*, an ancient duplicate of *cona*.** A. Unrooted protein tree of *cona* and its old duplicate *conta*; major lineages are labeled. Note, most of the *conta* branches in the *drosophila* subgenus are extremely short and barely visible. B. Annotation and germline expression of *conta* (purple) in *D. virilis* and the outgroup *S. lebanonensis*. C. Alphafold prediction of conta in representative species.

the outgroup, emerged prior to the last common ancestor of *Drosophila* and *Scaptodrosophila*. Unlike *cona*, *conta* sequence is conserved (Fig 4A), found in the same syntenic region (in the intron of the gene *teiresias;* S12 Fig), and highly expressed in the testes but not ovaries (Fig 4B). Structural prediction of *CONTA* reveals distinctly shorter coiled structures (Fig 4C). Given the absence of *conta*'s ortholog in *sophophora*–not even in the same syntenic location (S12 Fig)–it has likely been lost, further underscoring the propensity for SC paralogs to participate in germline function that may be evolutionarily fleeting.

## SC proteins are evolving under recurrent positive selection and accelerated by repeated duplications

Poor sequence homology can result from relaxed constraint due to reduced negative selection or adaptive protein evolution due to positive selection. Previously, Kursel et al. reported that the elevated rate of SC protein evolution in *Caenohabditis* reflect relaxed sequence constraint while the coiled-coil domains and protein lengths are both highly conserved [27]. However, Hemmer and Blumenstiel identified elevated rates of protein evolution and signatures of positive selection for *Drosophila* SC genes using both molecular evolution and population genetic metrics [14]. We reassessed the rates of protein evolution by estimating the branch-specific ratio of nonsynonymous and synonymous rates of protein evolution, represented by omega (also known as $D_N/D_S$). Values approaching 0 indicate negative selection while values close to or greater than 1 indicate relaxed constraint and positive selection, respectively [48]. Notably, gene-wide omega values, which are predominantly much lower than 1, are typically the composite of several modes of evolution as different residues and/or domains of the protein can be under different forms and levels of selection [49,50]. Our curated, species dense SC orthologs and paralogs enabled not only branch-specific, gene-wide estimates (Figs 5A, 5B and S13), but also detection of significant positive selection occurring only at portions of the protein coding sequence with the Hyphy package [51].

For the more conserved, lateral elements, despite the better preserved sequence homology and predominantly gene-wide omega of less than 1 (Figs 5A, S13 and Table 1), multiple branches still show signatures of positive selection; 19 out of 86 and 25 out of 100 branches

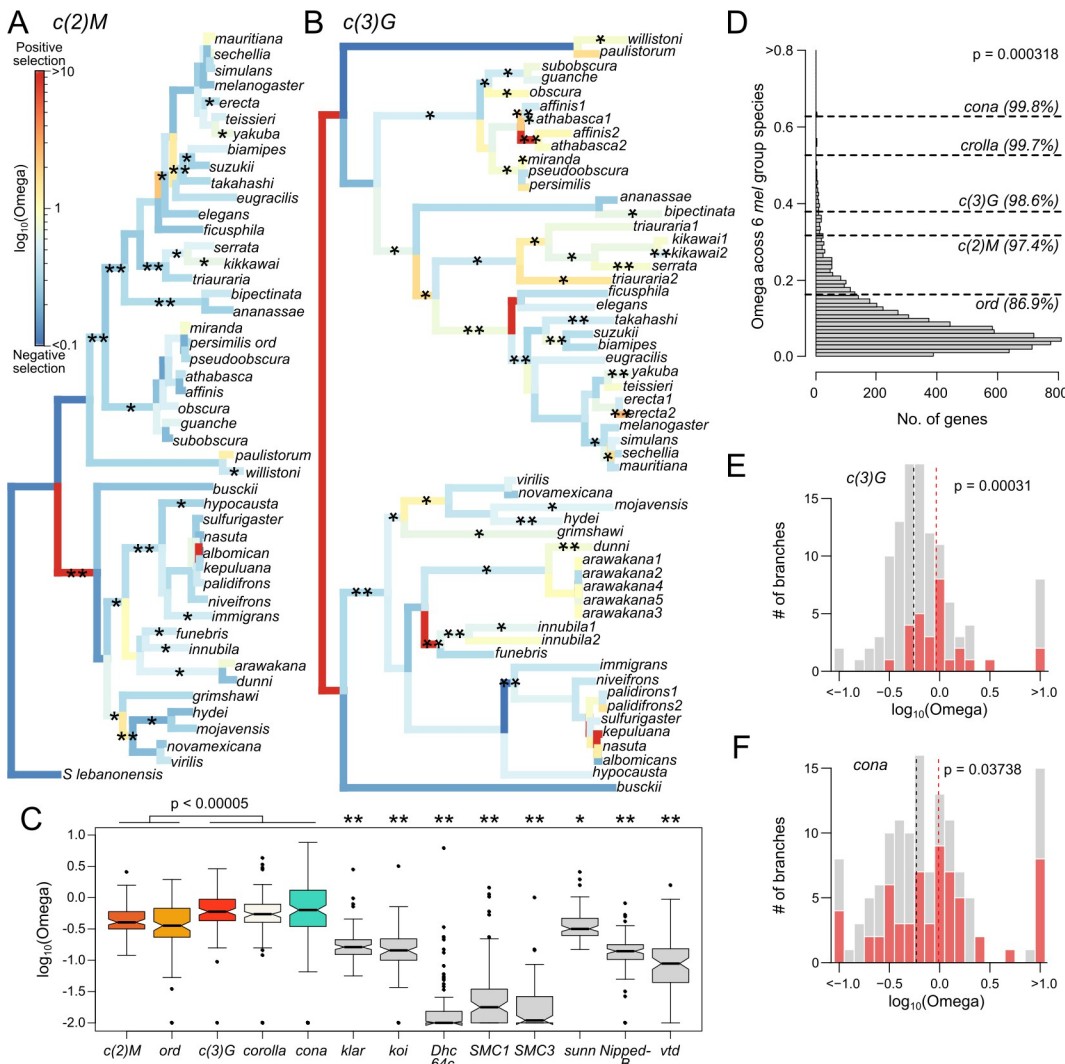

**Fig 5. Rate of protein evolution and signatures of positive selection of SC components.** A-B. Along the gene trees, branches are colored by their branch-specific rates of protein evolution (omega) with warmer colors representing higher omega. Branches inferred to have significant positive selection in part of the protein are labeled with asterisks (* = p < 0.05 and ** = p < 0.001). See S13 Fig for the remaining SC genes. C. Distribution of branch-specific omega values for the different SC components and genes required for pairing during early prophase I. P-values are from pairwise Wilcoxon's rank sum tests comparing between the lateral and central region genes. For non-SC genes, ** indicates significant differences (p < 0.005) when compared to all SC genes; * indicates significance comparisons except for ord. D. Omega of the SC genes compared to the genome-wide distribution estimated of by PAML for a subset of the species in the melanogaster group. E, F. Comparison of the distribution of omega values on branches following duplications (red) versus unduplicated branches (gray). Vertical dotted lines mark the median omega values for duplicated (red) and unduplicated branches (black). P-values were inferred from 1-tailed Wilcoxon's rank sum test.

display either gene-wide omega of greater than 1 or significant site-specific selection (nominal p < 0.05; we elected to use the nominal p-value as the test is demonstrably underpowered–see Materials and Methods) for *c(2)M* and *ord*, respectively. For the poorly conserved elements (Figs 5B, S12 and Table 1), not only do they have significantly higher omega than the lateral elements (p < 0.00005, pairwise Wilcoxon's rank sum tests; Fig 5C) indicative of higher rates of protein evolution, over 40% of the branches show either positive omega or significant signatures of positive selection at parts of the protein. Of note, we suspect the rate of protein

**Table 1. Branch specific rate of evolution of SC genes.**

| Genes | Number of branches | Branches with Omega > 1 | Median Omega | Significant branches (p < 0.05) | |
|---|---|---|---|---|---|
| | | | | Nominal P-value | FDR-adjusted P-value |
| *c(2)M* | 86 | 3 (3.5%) | 0.421 | 16 (18.6%) | 4 (4.7%) |
| *ord* | 100 | 15 (15.0%) | 0.353 | 11 (11.0%) | 3 (3.0%) |
| *c(3)G* | 106 | 28 (26.4%) | 0.584 | 30 (28.3%) | 13 (12.3%) |
| *corolla* | 86 | 15 (17.4%) | 0.54 | 27 (31.4%) | 22 (25.6%) |
| *cona* | 132 | 50 (37.9%) | 0.655 | 29 (22.0%) | 13 (9.8%) |

evolution for *c(3)G*, *corolla*, and *cona* may be underestimated due to poor amino acid alignments with many gaps across much of the protein as a result of rapid evolution. Collectively, the SC shows significantly elevated rates of protein evolution compared to the other genes involved in meiotic pairing; *sunn*, a gene that directly interacts with the SC, is the only exception with omega values similar to *ord* (Fig 5C), but nonetheless significantly lower than the other SC genes.

We further used PAML to infer the rate of protein evolution within the three well represented species groups and found evidence of group-specific positive selection although not significant in all instances (Table 2). For *c(2)M* and *ord*, the *melanogaster* and *obscura* groups, respectively, show significant group-wide positive selection based on the M7 vs. M8 model comparison, while in branch-specific tests, all species groups have at least one branch that is either significant or has omega > 1. For *c(3)G* and *corolla*, two of the three groups show significance. Lastly, while *cona* does not pass significance in any species group, they all show high numbers of branches with omega > 1. Because *cona* is a short gene of only ~220 AA, tests based on amino acid substitutions are inherently going to be underpowered. Similarly heterogeneous rates of protein evolution across clades have also been observed in rapidly evolving genes and pathways, such as those involved in programmed double strand breaks and DNA repair [52,53]. Restricting to a subset of 6 *melanogaster* group species with decent alignments,

**Table 2. Group specific PAML analyses.**

| | Species group | Models: No. of sequences | Branch-site No. of sig. branches | Free-ratio No. of branches w/ gene-wide Omega > 1 | Sites models | | | |
|---|---|---|---|---|---|---|---|---|
| | | | | | M7 v M8 (p-value) | M8a vs M8 (p-value) | M8%sites under positive selection | M8 Omega of selected sites |
| *c(2)M* | *mel* | 18 | 9 | 1 | 0.008252 | 0.281860 | 10.489 | 1.13326 |
| | *obs* | 8 | 0 | 1 | 1.000000 | 1.000000 | - | - |
| | *imm* | 8 | 1 | 3 | 0.251780 | 0.886410 | - | - |
| *ord* | *mel* | 18 | 9 | 2 | 0.298960 | 0.140460 | - | - |
| | *obs* | 13 | 2 | 8 | 0.000012 | 0.000108 | 5.638 | 3.20443 |
| | *imm* | 9 | 5 | 3 | 0.278730 | 0.170720 | - | - |
| *c(3)G* | *mel* | 21 | 10 | 5 | 0.000005 | 0.021788 | 9.181 | 1.34827 |
| | *obs* | 10 | 4 | 7 | 0.000000 | 0.000015 | 13.339 | 2.35341 |
| | *imm* | 9 | 5 | 5 | 0.255390 | 0.126480 | 2.852 | 6.26521 |
| *corolla* | *mel* | 19 | 17 | 5 | 0.000001 | 0.000034 | 7.929 | 1.8328 |
| | *obs* | 8 | 0 | 2 | 0.414690 | 0.630910 | - | - |
| | *imm* | 8 | 2 | 5 | 0.000001 | 0.000005 | 2.852 | 6.26521 |
| *cona* | *mel* | 26 | 7 | 12 | 0.095464 | 0.097430 | - | - |
| | *obs* | 13 | 0 | 6 | 0.671630 | 0.621020 | 34.992 | 1.00362 |
| | *imm* | 14 | 2 | 7 | 0.094451 | 0.239600 | 4.978 | 2.68505 |

we compared the omega values of SC genes to the genome-wide distribution and found that the SC genes fall between 86.9 and 99.8 percentiles and are significantly overrepresented with elevated values despite only 5 genes (Fig 5D; p = 0.000318 Wilcoxon's Rank Sum Test). Altogether these results demonstrate that all components of the SC have a history of recurrent adaptive evolution with the central region genes under frequent and repeated positive selection.

Copy number expansions can allow genes to diversify leading to new functions or subdivision of existing functions among the paralogs. Both scenarios are associated with elevated omega, either from relaxed functional constraint or positive selection for novel function. To test whether the recurrent duplications of SC components lead to elevated rates following functional diversification, we examined branches after duplications for *c(3)G* and *cona*. We found significantly elevated rates of protein evolution on such branches with median omega of 0.925 and 0.914, respectively, both significantly higher than single copy branches (Fig 5E and F; p = 0.000031 and 0.03788, respectively, one-tailed Wilcoxon's rank sum test). In particular, the *c(3)G2* paralogs with high testes expression in *D. triauraria*, *athabasca*, and *affinis* (Fig 3C–3E) all show clear signatures of adaptive evolution post duplication (Fig 5B). Although the terminal branches of the latter two species appear neutrally evolving, the parent branch displays highly elevated and significant omega, indicating strong positive selection in the common ancestor post duplication. Notably, the original copy, *c(3)G1*, in these cases also display signatures of positive selection suggesting that the duplications were followed by functional diversification to both. Moreover, other *c(3)G* duplicates, even the recent ones, show signatures of adaptive evolution including those in *D. erecta* and *kikkawai*. Similar for the *cona* duplicates, nearly all the branches of *cona2* in the serrata group show evidence of positive selection, which accounts for the overall longer branch lengths than those of *cona1* within the same group (S13 Fig). These repeated signatures of positive selection after duplications strongly argue for adaptive functional diversification, acting to further accelerate the already rapid evolution of the SC.

## Poor regulatory conservation of SC genes in both female and male germlines

With the exception of *ord* which maintains sister chromatid cohesion in the male germline [54,55], the *Drosophila* SC proteins are thought to primarily function in the ovaries and dispensable in the testes because males are achiasmic and mutant males do not show obvious meiotic or fertility defects [56–59]. However, a recent careful study of the male germline by Rubin et al. found that the progression of pre-meiotic chromosome pairing is slower in *cona* and *c(3)G* mutants, revealing putative male germline function despite the absence of SC assembly [60]. Inspired by these results, we examined the expression of SC genes in ovaries and testes RNA-seq datasets from a subset of 38 species (Fig 6A), 14 of which we generated and 24 curated from publicly available datasets (S3 Table). Other than *ord* which is testes-biased across all species, the expression of SC genes–even single copy ones–is highly unstable. Contrary to the naive expectation of high ovary and low testes activity, testes-biased expression appears to be the norm rather than the exception as SC components are more highly expressed in the testes in over 70% of the species. We further examined available testes single cell RNA-seq data of *D. miranda* [61], a species with high testes expression of *c(3)G*, and found expression concentrated in the pre-meiotic cell types such as germline stem cells and spermatogonia (S14 Fig). Expression of SC genes is also found in the early germline of *D. melanogaster* [62], consistent with their reported function in pre-meiotic pairing. Thus, even though they are primarily known for their role in female meiosis, SC genes can also have high testes expression. This is further supported by the heavily testes-biased expression of *corolla*, *c(2)M*, and *ord*, in the outgroup *S. lebanonensis*, suggesting ancestral testes expression.

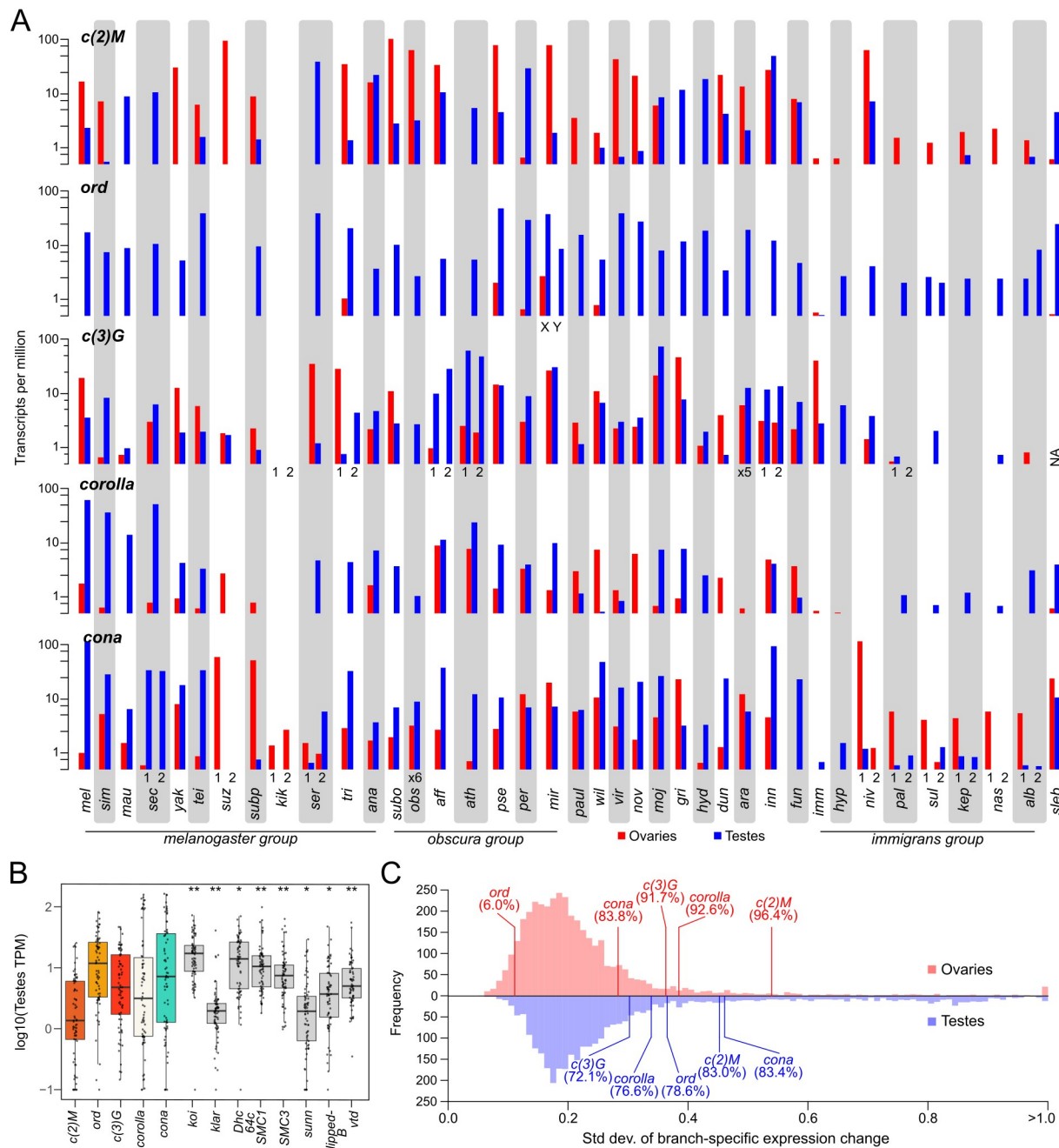

**Fig 6. Regulatory turnover of SC components in the testes and ovaries.** A. Ovaries (red) and testes (blue) expression (transcript per million in log scale) of SC genes across 38 species. Tandem duplicates with similar expression are collapsed into one. Other duplicates are labeled. Gray and white vertical bars differentiate between species. B. Distribution of testes gene expression across they phylogeny for SC genes (color) compared to meiotic pairing genes (gray); p-value is from the Brown–Forsythe test of equal variance [132] whereby the variances of two populations are compared and equal variance is rejected with a significant test; ** significant across all comparisons; * significant for some of the comparisons. For exact p-values of each pair-wise tests, see S15 Fig. C. Distribution of the standard deviation of the inferred branch-specific rate of regulatory evolution of SC genes and genes with identified orthology in Orthodb, with the percentile placement of SC genes shown. For the top panel, the distribution reflects the rate of regulatory evolution in the female germline inferred based on ovary expressed genes. Bottom panel depicts the rate distribution of testes genes.

Compared to the meiotic pairing genes which are active in overlapping ovarian cell types, SC genes have significantly elevated gene expression variance across species in both ovaries and testes (Figs 6B and S15). The striking lability of germline SC expression is particularly evident from several closely related species pairs whereby expression rapidly switches between testes- and ovaries-bias. For instance, *c(2)M* is ovaries-biased in *D. simulans* but testes-biased in the sister species *D. sechellia and mauritania* (Fig 6A). Similar rapid expression change of *c(2)M* is also observed in *D. pseudoobscura*/*D. persimilis* and *D. athatbasca*/*D. affinis* sister pairs, suggesting such regulatory turnover occurs frequently. As the RNA-seq datasets came from different studies, we tested the possibility that the expression differences resulted from external factors like growth conditions and examined SC expression from RNA-seq of studies where flies were reared in different environmental conditions. In the datasets examined [63–65], SC expression in the ovaries and testes shows minimal change in different rearing temperatures (S16 Fig). Additionally, drastic expression divergence of the germlines can be observed in RNA-seq datasets we generated of closely related species reared in the same controlled laboratory condition, including those in the *immigrans* group and the sister species *D. arawakana* and *D. dunni*. Therefore, highly variable expression of the SC across the genus likely reflects bona-fide regulatory divergence.

To determine whether the extent of regulatory turnover of SC genes is unusual among germline genes, we estimated the regulatory evolution across the phylogeny using the branch-specific gene expression changes (see Materials and Methods) and compared to the set of genes with clear orthology in OrthoDB [66]. Aside from *ord* which has consistently low expression in ovaries, the rates of regulatory changes of SC genes fall within the 83.8 and 96.4% percentiles of ovary-expressed genes (Fig 6C), revealing that they are indeed among genes with high regulatory turnover. For the testes, the regulatory variability of SC genes falls within the 72.2 and 83.4 percentile of testes-expressed genes (Fig 6C). While not as extreme as in ovaries, these values supports elevated rates of regulatory divergence, albeit among testes genes which tend to have higher regulatory divergence [67].

Other curious patterns include elevated testes but low-to-no ovary expression such as *c(3)G* in *D. obscura* and *D. hypocausta*. Most puzzling, there are multiple lineages where components of the SC show little-to-no expression in both gonads, such as *D. kikkawai* with little gonad expression of all components. The most extensive regulatory stability appears to be the *nasuta* subgroup where the expression of SC components are consistently low across the species, especially for *c(3)G* and *corolla*. These differences between species cannot be simply driven by differential tissue contribution in the dissections, as the expression of the SC components are poorly correlated, with many instances of high expressions of one component and low expression of others within the same sample. While it is tempting to infer function outside of the germline based on the absence of expression, even *D. melanogaster* ovaries show low expression of some of the SC components like *ord* and *cona* arguing that normal SC function may not require large transcript counts. Moreover, SC genes typically have low to no expression outside of the germline [62,68] (S17 Fig). Nevertheless, these patterns reveal that despite the essential roles in regulating crossovers, germline transcriptional regulation of many SC genes is labile and appears to be constantly evolving, echoing their protein and copy number evolution.

## Discussion

### Trials and tribulations of gene ortholog and CDS searches from genome releases

The species-dense examination of SC evolution was made possible by the large amounts of *Drosophila* genomes that have been recently published. One of the promises of the ever-

increasing number of genomes is to enable deep and broad investigation of the molecular evolution of genes and pathways. Naturally, analyses of genic evolution typically require alignments of full length CDSs, which are distilled from genome annotations. Since good gene structure inference requires additional data such as RNA-seq, only ~1/3 of the available *Drosophila* genomes have available annotations (including those we generated here), which were the focus of this study. Notably, recent studies were able to take advantage of more genomes by focusing on small genes such as protamines [69], sex peptide, and sex peptide receptor [70], which either have no intron or have simple gene structure, enabling protein construction without transcript information from RNA-seq. Even when we relied on annotations with RNA-seq support to identify gene orthologs from annotated genomes using reciprocal best blast hits, we were flummoxed by the regularity of annotation errors (S3 Table). Rapidly evolving genes may be particularly prone to such errors due to obscured sequence homology and, in the case of the SC genes, propensity to duplicate. Without careful manual curation, cross referencing with homology and RNA-seq data, annotations errors taken at face value would be falsely construed as evolutionary changes. Below, we discuss some of the pitfalls of identifying orthologs and paralogs from annotated genomes.

We find that NCBI annotations are generally more reliable than our in-house annotations with maker but has its own sets of issues, the biggest of which is the lack of access to the proprietary pipeline. Problems common to all pipelines include improper splice junctions producing truncated ORFs with premature stops and fused genes creating gene chimeras; tandem duplicates were particularly problematic for this as they cause extensive mapping errors of the RNA-seq reads between neighboring copies. These errors will cause protein sequences with excess length differences and can be detected as long stretches lacking homology between orthologs in multiple sequence alignments. The more insidious issues stem from genome assembly errors. Such errors most frequently occur as indels at homopolymers tracks, an issue common to Pacbio and Nanopore reads [71,72]. While released genomes should have been polished with Illumina reads to reduce such errors [71], we repeatedly identified annotation errors caused by frameshift indels at homopolymer tracks. The worst offender is a Refseq genome, released on and annotated by NCBI, that is rife with such indels (S18 Fig). In our in-house annotations with maker, such indels can cause expected frameshifts and truncated ORFs or unexpected splice junctions with no RNA-seq support. Puzzlingly, NCBI annotations add nonsensically short (<5 bp) introns around such frameshifts to maintain the ORF (S18 Fig). Because this creates near complete proteins but with small numbers of missing internal amino acids, it is only detectable by looking at variant calls in mapped Illumina reads. Careful examination of CDSs and protein sequences extracted from genome assemblies is therefore necessary to avoid erroneously calling species-specific substitutions, indels, nonsynonmous changes, psuedogenes, and splice variants.

## Mechanisms underlying recurrent duplications and rapid evolution of the SC

Our exhaustive survey for SC orthologs resulted in the surprising identification of many duplicates across *Drosophila*. Duplication is an important mechanism to diversify gene function as the resulting paralogs can either evolve novelty or compartmentalize existing functions, with reduced selective constraint on the gene as a single copy. Indeed, we find elevated rates of protein evolution following duplications indicative of both relaxed constraint and adaptive evolution. Many of the duplicates are young and species-specific showing no evidence of expression differences, and therefore unlikely to have diverged in function. For some of the older *c(3)G* and *cona* paralogs, we observed repeated and independent acquisition of distinct activities in

the male germline, such as testes-specific expressions and incorporation into lncRNA production. In the case of the tandem duplication of *c(3)G* in the ancestor of *D. affinis* and *D. athabasca* (Fig 3D-E), the two copies show clear signatures of positive selection driving diversified function evidenced by important structural differences (i.e. loss of terminal globular domains) in the protein and expression in different testes cell types. While divergence in expression domains is consistent with subfunctionalization, the adaptive protein evolution strongly argues for functional novelty. A parallel duplication *of c(3)G* occurred in *D. triauraria*, which also evolved into a testes-expressed paralog lacking the globular domains that interact with lateral elements. This striking molecular convergence raises the possibility of a common testes process acting to repeatedly drive the rapid evolution of SC proteins.

For reasons still unclear, the testes appear to be a unique regulatory environment in several ways. First it produces the largest repertoire of lncRNAs [73,74] and *de novo* genes [75–78], both of which can have critical roles in spermatogenesis. Considering our unexpected finding of frequent testes expression, we speculate that testes activity of SC genes–even single copy ones–may provide the opportunities to generate paralogs that can diversify. This path to functional novelty or diversification in the testes may be further facilitated by the elevated retrotransposon activity in spermatocytes [79] which can conceivably provide the necessary machineries for both retro- and tandem duplications. Our observation that *c(3)G* moved into a satellite-block in the lineages leading up to the *pseudoobscura* species subgroup (S8 Fig) lends support to this possibility. The second unique regulatory feature of the testes is that transcription seems to be promiscuous [80–82]. Therefore, high regulatory turnover or even elevated testes expression of SC may simply reflect "noisiness" of testes expression with no functional consequence. While we suspect this likely contributes to some of the regulatory changes, transcriptional promiscuity is unlikely to explain the observed protein sequence evolution, molecular convergence, and functional diversification after duplication. Indeed, in a thorough and careful genetic manipulation of young genes in the *Drosophila* testes, Kondo et al found widespread fitness consequences to the male germline [81]. Moreover, in the scenario where high testes transcription of SC genes has no direct downstream function, it nonetheless offers a substrate for selection and can facilitates the emergence of evolutionary innovations in the germline or other cell types [83,84].

Despite many paralogs, pseudogenization of duplicates also appears common. We have identified multiple instances where extant duplicates in one species have been lost in neighboring lineages (e.g. *c(3)G2* in *D. triauraria*, and *D. innubila*, and *conta*). We also found several examples of remnants of SC duplicates, including truncated copies of *c(3)G* and *corolla*. Such dynamic copy number changes raise a perplexing conundrum: why are *cona* and *c(3)G* prone to produce duplicates under positive selection only for the duplicates to end up as pseudogenes. One clue may come from the exceptional duplication of *ord* in *D. miranda*, where neo-X- and neo-Y-linked gametologs, along with other meiosis-related genes, have massively amplified in tandem [41]. The amplification is hypothesized to be the result of dosage-sensitive sex-ratio meiotic drivers precipitating an arms race for gene copy numbers on sex chromosomes [41,85]. Similar dynamics of repeated copy number evolution is also observed for sex ratio drivers and suppressors that manipulate DNA packaging in X- and Y- bearing sperms [69,86,87]. In such models of meiotic conflicts, temporary/young duplications may act to increase the gene dosage to either induce selfish transmission (such as biased sex ratio) or to act as suppressors of drive that restore fitness reduction associated with non-mendelian transmission. During oogenesis, SC assembly first begins at the centromere and persists there even after DSBs have been resolved and SC disassembled along the euchromatic arms. Analogously in the testes, *c(3)G* and *cona* have partial overlaps with the centromeres in the premeiotic cells where they seem to facilitate homolog pairing. Close associations with the centromeres and

kinetochores in either germlines raise the possibility that SC genes may be targets of meiotic cheating, or even cheaters themselves [88]. Once the conflict is resolved, drivers and suppressors may pseudogenize and degenerate as they no longer impart fitness benefits.

Since the intricate orchestration of chromosome movement is necessary for recombination and faithful disjunction, we find the regulatory volatility of SC genes in the ovaries to be perplexing, particularly in species where SC genes have little-to-no ovary expression. The repeated relocation of *c(3)G*, *cona*, and *corolla* to different chromosomal regions may confer some degree of regulatory change, but even *c(2)M*, which has remained in the same syntenic location, exhibits high regulatory turnover. It is tempting to speculate that the low ovary expression in multiples species corresponds to repeated loss of meiotic function, but we find other possibilities more likely. If SC proteins have long half-lives, minimal transcript production may be sufficient to support robust SC assembly. However, this possibility cannot address why species evolved to have drastically different expression profiles. While we examined available germline expression datasets under different environmental conditions and found minimal changes in SC expression, we cannot fully rule out extrinsic factors driving the expression lability of SC genes. Indeed, recombination rate is sensitive to environmental conditions and life history such as nutrition [89,90], temperature [4,91,92] and stresses [93], and age [94,95], and species can differ in their physiological responses that subsequently regulate SC activity. Such mechanisms can be beneficial in ensuring optimal recombination rates to modulate the amount of genotype diversity in the offspring [93] or proper progression of meiosis in suboptimal cellular conditions like extreme temperatures [11]. However, we note that the species for which we generated germline RNA-seq were raised under common laboratory conditions and were at standard, reproductively active age; yet we still observed drastic differences in SC regulation likely reflecting true regulatory divergence in both the male and female germlines.

In the ovaries, overexpression of central region genes causes abnormal SC polycomplexes which are associated with segregation defects and reduced fecundity [96,97]. In light of this female-specific deleterious potential at least in *D. melanogaster*, we suspect that intra-locus sexual antagonism may be at play [98]. Since the function of SC genes must differ in some way between female and male germlines as the latter lacks SC formation, they will have different dosage and fitness optima in the ovaries and testes. Gene expression is then likely to be evolutionarily unstable and suboptimal. Sexual antagonism can be resolved by the decoupling of sex-specific functions and selection [99], like when duplications occur, when sequence changes alter the molecular function and therefore fitness consequences of the gene, or when moved to or out of the sex chromosomes–all paths that have been observed for SC genes.

Our analyses of protein coding evolution demonstrate that SC genes have a complex evolutionary history with recurrent bouts of adaptation. The central elements and transverse filaments are particularly frequent targets of positive selection. In contrast, other genes necessary for chromosomal progression during early prophase are all conserved, other than *sunn* which directly interacts with the SC. While the elevated rates of evolution are likely driven in part by paralogs diversifying, orthologs without duplicates also show signatures of positive selection across the gene trees. This differs from the SC genes in *Caenorhabditis*, the protein sequence of which are poorly conserved but evolving neutrally [27]. Further, while *C(3)G* appears structurally conserved, the lengths of the proteins are far more variable with a coefficient of length variation 5 times higher than that of worms (0.17 vs. ~0.03 [27]). While this could reflect repeated adaptation in *Drosophila* female meiosis and meiotic recombination, our findings that SC genes is frequently active in testes where it is also ancestrally highly expressed compel us to consider additional avenues under recurrent positive selection, especially since spermatogenesis is fruitful grounds for meiotic conflicts, sexual antagonism, sexual selection, and molecular innovations. Pleiotropy like this tends to increase molecular constraint [100]. However, given

the sequence tolerance of the SC especially at the coiled-coil domains, dual or diverse function of *Drosophila* SC genes in both oogenesis and spermatogensis may instead predicate a curious scenario where positive selection in the latter causes protein changes that have little pleiotropic impact on the former. Dissecting the function and activity of SC genes in the *Drosophila* male germline, which is ironically achiasmate, will therefore be critical to understanding the diversity and evolution of meiotic recombination.

## Materials and methods

### High molecular weight DNA extraction and genome assembly

To assemble the genomes of *D. hypocausta* and *D. niveifrons*, we followed the Nanopore long read sequencing pipeline from [35,101]. In short, high molecular weight DNA was extracted using the Qiagen Blood & Cell Culture DNA Midi Kit with the spooling method from ~100 males of *D. hypocausta* strain 15115–1871.04 from the National Drosophila Species Stock Center and ~50 females of *D. niveifrons* strain LAE-276 from the Kyorin Drosophila Species Stock Center. DNA strands were hand spooled after precipitation, followed by gentle washing with supplied buffers.

### RNA-seq preparation and analyses

5 pairs of ovaries and testes were dissected from adult females and males and stored in Trizol at -80 degrees, followed by standard RNA extraction. RNA-seq libraries were generated using either the NEBNext RNA Library Prep Kit for Illumina with the Stranded and mRNA isolation Modules or the Illumina Truseq Stranded mRNA Library Prep kit. After quality check with the Fragment Analyzer at QB3-Berkeley, the libraries were sequenced by Novogene. We aligned the reads (both ones we generated and downloaded from SRA) using hisat2 (v2.2.1) [102] on either pair-end or single-end mode to their respective genomes with the–dta flag to allow for downstream transcriptome assembly. S2 Table lists the sources of the reads. After sorting the aligned reads with samtools (v1.5) [103], we used the featureCount (v2.0.3) in the Subread package [104] for read-counting over genes, allowing for non-uniquely mapped reads (-M flag). Read count tables were processed and analyzed in R (v4.2.2) and Rstudio (v2022.12.0). For gene expression analyses, we normalized the read counts across samples by converting them to transcript per million (TPM) [105]. For species where we needed to do de novo gene annotation, we used stringtie v2.1.6 [106] on default for genome-guided transcript assembly.

### Gene annotation and manual curation of gene structures

For species that required gene annotation, we ran three rounds of maker [107]. For evidence-based ab initio gene prediction in the first round, we supplied the transcript assembly from stringtie, de novo repeat index from RepeatModeler2 [108], transcript sequence from closely related-species and protein sequence data from *D. melanogaster* and *D. virilis* downloaded from FlyBase. The maker results from round one were used to train the species specific gene model using SNAP [109]. The resulting snap.hmm file was fed back into maker for round 2. We iterated this process again, refining the gene models for a 3rd round of maker.

For malformed or missing annotations, we first visualized the gene structures and RNA-seq reads mapping around them using IGV (v2.16.0) [110]. Additionally, we manually defined the region of the genome showing gene homology by blastn-ing the well-formed ortholog from a closely related species to the genome. The combination of RNA-seq reads mapping and the blast-hit boundaries provided evidence to correct erroneous exon-intron injunctions,

truncated annotations, chimeric gene structures, and absent annotations. To update the annotations file (.gff file), we used the either the genome browser GenomeView (v2250) [111] to manually edit or add the gene structures including mRNAs, exons, CDSs, or GenomeThreader [112] (v 1.7.3, -gff3out flag) to predict the gene structure based on the manually constructed CDS. All edited genes have at least full open reading frames, although 5' and 3' UTRs may be missing. All manually annotated features were marked by the flag "hand" in the gffs. The updated.gff is then exported and sorted using GFF3sort [113], and transcript sequences are retrieved using gffread (0.9.12) [114]. In several instances, we noticed assembly errors leading to malformed genes. One was *ord* in D. nasuta which had a stretch of N's within the gene body indicating scaffolding points. The other was in D. *neivifrons* where c(3)G was annotated as two fragments. This was due to a deletion of a single nucleotide in the genome causing a shifted reading frame which led to malformed annotations. The deletion was revealed by RNA-seq read mapping, whereby all reads showed a one basepair insertion. We fused the fragmented annotations into one and corrected the transcript sequence to rectify the erroneous deletion. Lastly, we initially could not identify *corolla* in the primary NCBI genome assembly of *D. funebris* strongly suggesting gene is loss; however, we were subsequently able to identify it in an unplaced repeat-rich contig in a separate assembly. For species with malformed annotations that had a closely related species with well-formed orthologs, we used the program Liftoff [115] to convert the annotation from one genome to the other.

## Homolog search with reciprocal best blast hits of transcripts and/or coding sequences

To identify orthologs and paralogs using a reciprocal best blast hit strategy, we reciprocally blastn-ed transcript sequences from species pairs using the commands:

blastn -task blastn -query species1.transcripts -db species2.transcripts -outfmt "6 qseqid sseqid pident length qlen slen mismatch gapopen qstart qend sstart send evalue bitscore" -evalue 1

blastn -task blastn -query species2.transcripts -db species1.transcripts -outfmt "6 qseqid sseqid pident length qlen slen mismatch gapopen qstart qend sstart send evalue bitscore" -evalue 1.

For publicly available genomes with annotation files, we generated the transcript sequences using gffread, otherwise we used transcript sequences generated by maker. We then used grep to identify the blast hits and checked whether they are reciprocal best hits of each other. For the *Sophophora* and *Drosophila* sub-genera, we used *D. melanogaster* and *D. virilis* sequences downloaded from Flybase as the focal species and blasted them first to their close relatives. When one species yields no blast hit for a gene, we then use other closely related species where the orthologs was successfully identified. If no hits can be identified for a species or a clade, we then repeat the same procedure using tblastn to identify translated protein sequences, as amino acid can be more conserved than nucleotide sequence. If tblastn fails to identify a homologous transcript, we then blastn-ed to the genome sequence. True absences/loss of a gene will yield poor or no blast hits, while missing annotation will result in clear noncontiguous hits with gaps corresponding to introns.

## Microsynteny surrounding homologs and chromosome placement

We extracted the sequences of the homologs including 50kb up and downstream using bedtools slop and bedtools getfasta. We then pairwise blastn-ed the sequences to each other and filtered out alignments with E-values of < 0.01 or shorter than 100 bp. To infer the extent of homology in the flanking sequences, we calculated the proportion of sequence aligned, excluding the positions of the homolog. Genes are deemed to be in non-syntenic regions if they share < 5% flanking homology. For genomes without Muller element designation of

chromosomes, we assigned Muller elements by aligning to the genome of a closely related species with minimap2 [116] where the Muller elements have been determined.

## Phylogeny construction

We retrieved the CDS for all genes, removed the stop codon, and converted them first to protein sequences using EMBOSS Transeq [117]. We then aligned the protein sequences using three aligners with the commands: prank (v.170427) -protein -showtree [118], mafft (v7.505) —localpair—maxiterate 1000 [119], and muscle (v5.1) [120]. The resulting multi-sequence alignment fasta file were used as the input for iqtree (v1.6.12) [121] with the flags -AA and -bb 2000 for 2000 iterations of ultrafast boot-strapping [122]. These trees were manually rooted with *S. lebanonensis* as the outgroup species in FigTree (v1.4.4) [123], and then Node labels added with phytools (v1.5.1) [124] to the trees to facilitate downstream rate of evolution analyses with Hyphy. We then selected the resulting trees with the best bootstrap support and concordance with species tree. Note, the *c(3)G* trees could not be rooted due to the absence of ortholog in the outgroup and *cona* trees were highly inconsistent across alignment methods with many poorly supported nodes.

## Rate of protein evolution and positive selection with HyPhy and PAML

We used TranslatorX [125] to align the CDS sequence based on the protein alignments. Providing the CDS alignments and the protein trees, we used the ABSREL module in HyPhy (v2.5.51) [51,126] to infer the branch-specific rate of protein evolution (Omega) and significant signatures of positive selection. We wrote a custom script (github.com/weikevinhc/phyloparse) to parse the HyPhy.json output in R where the trees were reoriented with phytools and visualized with colors representing gene-wide omega values. The nominal p-values were used for significance as the statistical procedure appears underpowered (S19 Fig). In addition, we used PAML (v4.9j) [127] for the species-group specific test of recurrent positive selection. From the genus wide alignment of SC genes, we extracted six species, *D. melanogaster*, *D. simulans*, *D. sechellia*, *D. yakuba*, *D. erecta, and D. ananassae*, the same six *Drosophila* species used to calculate dN/dS previously in Clark et al 2007 [128]. We then constructed maximum-likelihood trees using iqtree using parameters '-m MFP -nt AUTO -alrt 1000 -bb 1000 -bnni'. We then used the PAML parameter (model = 0 and CodonFreq = 2) to estimate the rate of protein evolution across the six species. For PAML analyses of each species group, we extracted the species from the genus-wide alignments to avoid realignment and removed gaps and trimmed the tree to the relevant species to avoid realignments and tree construction. We then used the wrapper software package PaPAML for branch-site models [129,130] and PAML (model = 1, CodonFreq = 2) for the site and free-ratio models.

## Protein structure prediction with AlphaFold

Structures of proteins previously annotated in NCBI were retrieved from the AlphaFold Protein Structure Database [44]. For genes we annotated, we used ColabFold (v1.5.2), an implementation of AlphaFold on the Google Colab platform [45] and selected num_recycles 24, producing structure predictions that were visualized in UCSF ChimeraX [131].

## Branch-specific regulatory evolution

*Drosophila*-specific orthology data was downloaded from https://www.orthodb.org/ [66] and we trimmed the species to only those that have RNA-seq data resulting in 15 species (*mel, sim, yak, tei, suz, subp, kik, ana, obs, mir, vir, hyd, nov, inn, alb*), and then removed genes with

missing orthology in more than one species producing a set of 9358 genes. To infer the rate of regulatory evolution, we used the fastAnc() function in phytools [124] to reconstruct the ancestral state at internal nodes with either log-transformed ovary or testes expression (in TPM) as the "trait" across the phylogeny. The rate of regulatory change at each branch is then the difference between parent node and children nodes/tips.

## Supporting information

**S1 Table. Overview of datasets.**
(XLSX)

**S2 Table. New genome assemblies and annotations.**
(XLSX)

**S3 Table. SC ortholog and paralog identification.**
(XLSX)

**S1 Fig. IGV genome tracks showing examples of common sources of annotation errors.**
RNA-seq read depth, splice junctions, and reads are shown. Erroneous and corrected annotations are the two bottom tracks, respectively. A. Absent annotations despite RNA-support (circled by a red box). B. Neighboring genes are fused into a long chimeric gene. C. Annotation error due to assembly errors that cause frameshifts. RNA-seq reads mapping all show a 1bp insertions (purple marks on the reads and circled by red boxes) indicating that the assembly has misassembled these regions introducing two 1bp deletions. Such errors in the exons destroys the ORF leading to erroneous inferences.
(PDF)

**S2 Fig.** Dotplots of self-alignments of the 500kb regions on the neo-X (A) and neo-Y (B) containing ord. Repeated tandem duplications can be observed generating many alignments off the diagonal.
(PDF)

**S3 Fig. Sequence similarity of orthologs of genes involved in meiotic pairing during female meiosis as measured by blast (same as Fig 1D–1F).** Pairwise % identity values above and below the diagonal were estimated based on blastn of the CDS and blastp of the AA sequences, respectively.
(PDF)

**S4 Fig.** Gene trees for c(2)M (A), ord (B), corolla (C) and cona (D-F) constructed using the protein alignments. Trees for cona are based on Prank (D), MUSCLE (E), and MAFFT (F) protein alignments.
(PDF)

**S5 Fig. Dotplot showing the alignment between subobscura and obscura.** 6 tandem duplicates of cona in D. obscura can be observed.
(PDF)

**S6 Fig. Muller elements in which SC components are found are labeled.** Each arrow indicates a syntenic region where orthologs are found. Muller elements are labeled by different colors and are not drawn to scale. Order of the arrows do not reflect their relative chromosomal locations. Open arrow for corolla indicates insertion into repeat rich pericentromeric regions. B. Dotplot of the genomic region surround c(3)G paralogs in D. innubila compared to D. funebris. Phylogenetic reconstruction of c(3)G position and movements in the genome. Color of branches indicate the Muller elements in which c(3)G resides. Different patterns represent

different, non-syntenic locations, on the Muller elements. Arrows point to the D. innubila duplicates which are found in ancestral and derived regions suggesting an old duplication prior to species split with D. funebris.
(PDF)

**S7 Fig. Truncations of tandem copies of corolla in D. arawakana.** A. Alignment of corolla CDS (center track) to the D. dunni (top track) and D.arawakana (bottom track) genomes. B. Self alignment of the genomic region containing corolla revealing complex tandem repeat structures.
(PDF)

**S8 Fig. c(3)G movement in the pseudoobscura subgroup.** A. c(3)G moved to Muller E in the pseudoobscura subgroup. Blastn alignment between the c(3)G location in D. miranda to the syntenic region in D. affinis which lacks c(3)G. c(3)G and flanking genes are boxed in red. B. Self alignment of the region c(3)G migrated to show extensive tandem repeat structure in D. affinis. Region where c(3)G inserted into boxed in red. C. Self alignment of the syntenic region in D. obscura.
(PDF)

**S9 Fig. Coiled-coil domain prediction for representative c(3)G orthologs and paralogs.**
(PDF)

**S10 Fig. Expression of c(3)G1 and c(3)G2 in D. affinis single nuclei RNA-seq.**
(PDF)

**S11 Fig. IGV genome tracks of D. serrata showing expression surrounding cona2 (marked by purple reads) and the anti-sense lncRNA (marked by pink reads).** Pink and purple differentiate reads originating from the sense and antisense strands. Note the genome track is in the reverse orientation compared to Fig 3D.
(PDF)

**S12 Fig. Conta location in the D. virilis genome.** It is embedded in the intron of the gene teiresias. Note the lack of ortholog in the gray track.
(PDF)

**S13 Fig. Branch specific Ka/Ks for ord, corolla, and cona.**
(PDF)

**S14 Fig. Expression of c(3)G in testes single cell RNA-seq data for D. miranda.** UMAP projection of testes cell types. Relevant cell types are labeled.
(PDF)

**S15 Fig. Gene expression of SC and pairing genes in different Drosophila species.** A. Ovary expression. P-value of pairwise Brown Forsythe test of equal variance for ovary expssion (B) and testes expression (C).
(PDF)

**S16 Fig.** Expression of SC genes in different temperatures in D. melanogaster ovaries (A), D melanogaster males whole bodies (B), and D. suzukii testes (C).
(PDF)

**S17 Fig. SC gene expression across tissues.** Modified from the adult cell type atlas from Flybase.
(PDF)

**S18 Fig. Repeated indels at homopolymer tracks in the Refseq D. suzukii genome cause short introns in exons in NCBI annotations.**
(PDF)

**S19 Fig. Power analysis of Hyphy's statistical procedure to detect significant positive selection.** Qunatile quantile plot of the observed and expected p-value (in -log10 scale) for the rapidly evolving c(3)G and conserved vtd shows that in both cases the reported p-values are not sensitive especially when p-values are high. Despite this, c(3)G clearly deviates from expected due to elevated rates of protein evolution, but multiple testing correction removes large number of the nominally significant data points.
(PDF)

## Author Contributions

**Conceptualization:** Kevin H-C. Wei.

**Data curation:** Rana Zakerzade, Ching-Ho Chang, Kamalakar Chatla, Ananya Krishnapura, Samuel P. Appiah, Jacki Zhang, Robert L. Unckless, Justin P. Blumenstiel, Kevin H-C. Wei.

**Formal analysis:** Rana Zakerzade, Ching-Ho Chang, Robert L. Unckless, Kevin H-C. Wei.

**Funding acquisition:** Kevin H-C. Wei.

**Investigation:** Rana Zakerzade, Justin P. Blumenstiel, Kevin H-C. Wei.

**Methodology:** Kevin H-C. Wei.

**Project administration:** Kevin H-C. Wei.

**Resources:** Kamalakar Chatla, Robert L. Unckless, Doris Bachtrog, Kevin H-C. Wei.

**Software:** Kevin H-C. Wei.

**Supervision:** Doris Bachtrog, Kevin H-C. Wei.

**Validation:** Kevin H-C. Wei.

**Visualization:** Robert L. Unckless, Kevin H-C. Wei.

**Writing – original draft:** Kevin H-C. Wei.

**Writing – review & editing:** Ananya Krishnapura, Robert L. Unckless, Justin P. Blumenstiel, Kevin H-C. Wei.

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
