## [Decision Letter · Decision Letter 0]

3 Sep 2024

Dear Dr Wei,

Thank you very much for submitting your Research Article entitled 'Diversification and recurrent adaptation of the synaptonemal complex in Drosophila' to PLOS Genetics.

The manuscript was fully evaluated at the editorial level and by independent peer reviewers. The reviewers appreciated the attention to an important problem, but raised some substantial concerns about the current manuscript. Based on the reviews, we will not be able to accept this version of the manuscript, but we would be willing to review a much-revised version. We cannot, of course, promise publication at that time.

If you decide to revise the manuscript for further consideration at PLOS Genetics, please aim to resubmit within the next 60 days, unless it will take extra time to address the concerns of the reviewers, in which case we would appreciate an expected resubmission date by email to plosgenetics@plos.org.

To resubmit, log into your Editorial Manager account and select the option 'Revise Submission' in the 'Submissions Needing Revision' folder.

We are sorry that we cannot be more positive about your manuscript at this stage. Please do not hesitate to contact us if you have any concerns or questions.

Yours sincerely,

Colin Meiklejohn

Academic Editor

PLOS Genetics

Justin Fay

Section Editor

PLOS Genetics

Dear Dr. Wei,

Thank you very much for submitting your manuscript "Diversification and recurrent adaptation of the synaptonemal complex in Drosophila”" (PGENETICS-D-24-00785) for consideration at PLOS Genetics. Your manuscript has been evaluated by three reviewers, all of whom recognized the novel and important contributions your team has made. However, all the reviewers also raised points that need to be addressed before your manuscript would be considered for publication. I have summarized below two main comments that were identified by more than one reviewer:

1. The manuscript reports that structural and expression evolution within synaptonemal complex genes is unusually rapid, but lacks explicit contrasts with other “control” gene sets to support this characterization. The comparison of Blastp % identity with genes involved in meiotic pairing is very informative regarding rates of protein evolution; can an analogous comparison be made to determine that the structural (gene duplication) and regulatory evolution of SC genes is also unusual? Given the effort and care that was required to uncover the patterns of SC gene evolution in this study, are the authors confident that similar evolutionary dynamism would not be discovered in other multi-protein complexes following such careful annotation across this phylogenetic range?

2. Some of the manuscript’s conclusions assume gene function from gene expression data. This may be generally problematic, but is likely to be even more so for gene expression in reproductive tissues, particularly the testes. One concrete example is the disconnect between expression of SC genes and the phenotypes associated with mutant alleles of these genes in D. melanogaster. While this specific paradox is noted in the manuscript, the wider implications of inferring function from gene expression in species where no mutant data is available could be more carefully considered.

A revised manuscript should also consider individual reviewer comments. In particular, the discrepancies between the raw data and the results presented in the body of the manuscript, and the gaps in details of the PAML analyses that were noted by Reviewer 2 need to be addressed.

Reviewer's Responses to Questions

**Comments to the Authors:**

Reviewer #1: Review is uploaded as an attachment

Reviewer #2: my review is uploaded as an attachment

Reviewer #3: Zakerzade and Chang et al. take advantage of the plethora of new Drosophilidae sequence data and new data they generate to analyze the patterns of evolution of genes in the synaptonemal complex (SC). Through a painstaking process, they demonstrate that despite being essential for female meiosis, genes in the synaptonemal complex are rapidly evolving in protein sequence, copy-number, and regulation. They show that SC genes are prone to frequent duplication across Drosophilidae, and that these paralogs often follow a similar pattern of structural evolution. Perhaps most interestingly, the authors find that both single copy SC genes and their paralogs are prone to acquiring testes expression. This is strange as meiotic pairing in the Drosophila male germline is achiasmate and doesn’t use the synaptonemal complex. This tendency thus indicates that SC genes and their paralogs may play an important and largely uncharacterized role in the male germline.

This work constitutes a major step forward in our understanding of the evolution of the synaptonemal complex. It also sets the stage for future functional studies to better understand the mysterious role that the SC genes may play in the male germline in D. melanogaster and across Drosophila. Zakerzade and Chang et al. also point out a crucial hurdle for evolutionary genetic studies in Drosophila – gene annotation. They show that despite the hundreds of high-quality Drosophila genomes now available, proper annotation is the rate limiting factor for analyses of rapidly evolving genes, and manual annotation is often still needed.

There are a few aspects of this manuscript that can be improved. Namely, the authors rely on expression data to support the claim that SC genes and their paralogs may have a critical function in the male germline. This may lead to some potentially faulty assumptions. For example, looking at expression alone, the pattern of expression of C(3)G and cona in D. melanogaster (high in testes, low in ovaries) would suggest that it should play a crucial role in male germline development, and a less important role in females. This doesn’t seem to be the case. The Rubin et al. (PNAS, 2022) study they cite shows that though they suffer a reduction in efficiency of premeiotic pairing, C(3)G and cona mutant males do not show any defects in chromosome territory formation, chromosome segregation or reduced fertility. The authors should consider discussing explanations for the discrepancy between gene expression data and experimental observations of necessity in D. melanogaster, as this point is crucial for the conclusions they make about a potential male germline function of SC genes across Drosophila.

The authors never discuss the expression patterns of SC genes outside of the male and female germline. I wonder if they also see somatic regulatory variability for SC genes, or whether their variability in expression is limited to the germline.

Smaller edits:

Sentence ending on line 57 is missing a reference.

Line 232: Figure 2 C-E seem to have no information on the expression of c(3)G paralogs.

Line 272: It is not immediately clear how Figure 4A demonstrates conservation. It may be that the gene tree is much tighter, but this is hard to see visually.

Line 276: The wrong figure is referenced.

Line 293: Supplementary figure 12 seems to be the wrong figure.

Figure 5 legend: Supplementary figure 12 seems to be the wrong figure.

Line 332: Supplementary figure 12 seems to be the wrong figure.

Line 418: Supplementary figure 15 seems to be the wrong figure.

Line 449-450: The authors use ‘pseudoobscura subspecies.’ This may be misleading, as there are multiple subspecies of Drosophila pseudoobscura, but they are intending to refer a translocation that occurred in the ancestor of D. pseudoobscura, D. persimilis, and D. miranda. Perhaps they should change to ‘pseudoobscura group species.’

Perhaps a more comprehensive figure legend would be helpful for supplementary figure 2.

Supplementary figure 5 legend is cut off.

**Have all data underlying the figures and results presented in the manuscript been provided?**

Reviewer #1: Yes

Reviewer #2: None

Reviewer #3: Yes

PLOS authors have the option to publish the peer review history of their article (what does this mean?). If published, this will include your full peer review and any attached files.

Reviewer #1: No

Reviewer #2: No

Reviewer #3: No

---

## [Decision Letter · Decision Letter 1]

26 Nov 2024

PGENETICS-D-24-00785R1Diversification and recurrent adaptation of the synaptonemal complex in DrosophilaPLOS Genetics

Dear Dr. Wei, Thank you for submitting your manuscript to PLOS Genetics. After careful consideration, we feel that it has merit but does not fully meet PLOS Genetics's publication criteria as it currently stands. Therefore, we invite you to submit a revised version of the manuscript that addresses the points raised during the review process. Please submit your revised manuscript within 30 days Dec 26 2024 11:59PM. If you will need more time than this to complete your revisions, please reply to this message or contact the journal office at plosgenetics@plos.org. Please include the following items when submitting your revised manuscript: *
A rebuttal letter that responds to each point raised by the editor and reviewer(s). You should upload this letter as a separate file labeled 'Response to Reviewers'. This file does not need to include responses to formatting updates and technical items listed in the 'Journal Requirements' section below. *
A marked-up copy of your manuscript that highlights changes made to the original version. You should upload this as a separate file labeled 'Revised Manuscript with Track Changes'. *
An unmarked version of your revised paper without tracked changes. You should upload this as a separate file labeled 'Manuscript'.

We look forward to receiving your revised manuscript. Kind regards,Colin MeiklejohnAcademic EditorPLOS Genetics

Justin Fay

Section Editor

PLOS Genetics

Aimée Dudley

Editor-in-Chief

PLOS Genetics

Anne Goriely

Editor-in-ChiefPLOS Genetics

**Additional Editor Comments:**

Dear Dr. Wei,

Thank you very much for submitting your revised manuscript "Diversification and recurrent adaptation of the synaptonemal complex in Drosophila”" (PGENETICS-D-24-00785R1) to PLOS Genetics. Your revised manuscript was evaluated by the original three reviewers, and reviewers 1 & 3 were satisfied with the changes. Reviewer #2 also felt that the revised manuscript was greatly improved, but has a few more minor recommendations they feel are important before publication. I see that the main text now does mention the issue with nominal p-values from the HyPhy test, so I don’t think that needs to be addressed (perhaps the reviewer missed that revision). I agree with the reviewer that the use of “consistently” to describe the PAML results (line 331 in the revision) does not reflect the set of P-values in Supplemental Table 4, and that it would be beneficial to move this table to the main text. I also agree that adding the M8 vs. M8a comparison from PAML would be a valuable addition to the analyses. Finally, I also agree that in the instances indicated by the reviewer, “expression” is a better word choice than “activity”. I personally also agree that some of the adjectives/adverbs (curiously, surprisingly) are not necessary, but I believe that should be the authors’ decision.

I had one question regarding Figure 6B. The manuscript text indicates that the expression variance is greater for SC genes than meiotic pairing genes, but unless I misunderstand the Brown–Forsythe test, I assume that the greater variance in SC genes would result in significant P-values for those genes? In Figure 6B the ** indicate it is the pairing genes with significantly different variances. Given that the Brown–Forsythe test is not so commonly used, I think some explanation here might help the reader, particularly if a significant result indicates equal variance.

I also found a number of little typos and errors, but I probably missed others, so I recommend a careful re-reading of the manuscript before resubmitting.

page 8 line 168: “should be previously unaccounted for”

line 237: C(3)G should be c(3)G

line 309: Notedly should be Notably

line 325: what is the word “Note” doing here?

line 489: “protein sequences” should be “protein sequence”  **Journal Requirements:**

Please update "Summary" section to be "Author Summary". This should appear in your manuscript between the Abstract and the Introduction, and should be 150u2013200 words long. The aim should be to make your findings accessible to a wide audience that includes both scientists and non-scientists. Sample summaries can be found on our website under Submission Guidelines:

https://journals.plos.org/plosgenetics/s/submission-guidelines#loc-parts-of-a-submission

**Reviewers' comments:**

Reviewer's Responses to Questions

**Comments to the Authors:**

Reviewer #1: The revised manuscript is significantly improved and more straightforward to follow than the previous version. I appreciate the authors’ engagement with my comments, particularly regarding enhancing figure clarity. The expanded analysis of regulatory evolution across a larger set of species adds robustness to their findings and highlights the distinct evolutionary dynamics of SC genes. Additionally, the revised discussion also provides a more well-rounded exploration of the mechanisms driving SC diversification, adding depth to the manuscript’s insights. Finally, the revised language regarding gene expression in testes and the nuanced discussion of functional implications strengthen the manuscript by clarifying the limits of their conclusions. Overall, I commend the authors for their improved manuscript and look forward to seeing further investigations into the functional consequences of adaptive evolution at SC genes.

Reviewer #2: My review has been uploaded as an attachement.

Reviewer #3: As previously stated, this work provides valuable insight into the evolutionary pattern of synaptonemal complex genes and points out a possible widespread function in the male germline. The authors addressed my largest hold-up by changing their language when inferring gene function from testes expression and adding analysis in the discussion addressing this matter. I am satisfied with the changes made by the authors. This paper is now more transparent, and easier for the reader to interpret.

**Have all data underlying the figures and results presented in the manuscript been provided?**

Reviewer #1: Yes

Reviewer #2: Yes

Reviewer #3: Yes

PLOS authors have the option to publish the peer review history of their article (what does this mean?). If published, this will include your full peer review and any attached files.

Reviewer #1: No

Reviewer #2: No

Reviewer #3: No

**Figure resubmission:** While revising your submission, please upload your figure files to the Preflight Analysis and Conversion Engine (PACE) digital diagnostic tool, https://pacev2.apexcovantage.com/. PACE helps ensure that figures meet PLOS requirements. To use PACE, you must first register as a user. Registration is free. Then, login and navigate to the UPLOAD tab, where you will find detailed instructions on how to use the tool. If you encounter any issues or have any questions when using PACE, please email PLOS at figures@plos.org. Please note that Supporting Information files do not need this step. If there are other versions of figure files still present in your submission file inventory at resubmission, please replace them with the PACE-processed versions.  **Reproducibility:** To enhance the reproducibility of your results, we recommend that authors deposit laboratory protocols in protocols.io, where a protocol can be assigned its own identifier (DOI) such that it can be cited independently in the future. Additionally, PLOS ONE offers an option to publish peer-reviewed clinical study protocols. Read more information on sharing protocols at https://plos.org/protocols?utm_medium=editorial-email&utm_source=authorletters&utm_campaign=protocols

---

## [Decision Letter · Decision Letter 2]

19 Dec 2024

Dear Dr Wei,

We are pleased to inform you that your manuscript entitled "Diversification and recurrent adaptation of the synaptonemal complex in Drosophila" has been editorially accepted for publication in PLOS Genetics. Congratulations!

Yours sincerely,

Colin Meiklejohn

Academic Editor

PLOS Genetics

Justin Fay

Section Editor

PLOS Genetics

Aimée Dudley

Editor-in-Chief

PLOS Genetics

Anne Goriely

Editor-in-Chief

PLOS Genetics

Comments from the reviewers (if applicable):

Reviewer's Responses to Questions

**Comments to the Authors:**

Reviewer #2: The authors have now added results from M8a vs M8 and displayed all the PAML results in Table 2. In one instance, c(2)m in the melanogaster group, the M7vs M8 sites model result is strongly significant but the M8a vs M8 result is not. One possible explanation is that divergence in c(2)m is primarily driven by neutral evolution. The authors could mention this if they choose.

My remaining suggestions were mostly a matter of preference in writing style. I prefer conservative language in scientific writing because the alternative reads as if the authors have a desired outcome. I am satisfied that the authors made some of my suggested changes.

**Have all data underlying the figures and results presented in the manuscript been provided?**

Reviewer #2: Yes

PLOS authors have the option to publish the peer review history of their article (what does this mean?). If published, this will include your full peer review and any attached files.

Reviewer #2: No

**Data Deposition**

http://datadryad.org/submit?journalID=pgenetics&manu=PGENETICS-D-24-00785R2

**Press Queries**

---

## [Editor Report · Acceptance letter]

30 Dec 2024

PGENETICS-D-24-00785R2 

Diversification and recurrent adaptation of the synaptonemal complex in *Drosophila*

Dear Dr Wei, 

We are pleased to inform you that your manuscript entitled "Diversification and recurrent adaptation of the synaptonemal complex in *Drosophila*" has been formally accepted for publication in PLOS Genetics! Your manuscript is now with our production department and you will be notified of the publication date in due course.

With kind regards,

Katalin Szabo

PLOS Genetics

On behalf of:
